# Mechanistic Insights in Hemophagocytic Lymphohistiocytosis: Subsequent Acute Hepatic Failure in a Multiple Myeloma Patient Following Therapy with Ixazomib-Lenalidomide-Dexamethasone

**DOI:** 10.3390/jpm12050678

**Published:** 2022-04-23

**Authors:** Catalin Constantinescu, Bobe Petrushev, Ioana Rus, Horia Stefanescu, Otilia Frasinariu, Simona Margarit, Delia Dima, Ciprian Tomuleasa

**Affiliations:** 1Department of Hematology, Iuliu Hatieganu University of Medicine and Pharmacy, 400349 Cluj-Napoca, Romania; constantinescu.catalin@ymail.com; 2Department of Anesthesia and Intensive Care, Iuliu Hatieganu University of Medicine and Pharmacy, 400349 Cluj-Napoca, Romania; margaritsim@yahoo.com; 3Intensive Care Unit, Emergency Hospital, 400006 Cluj-Napoca, Romania; 4Medfuture Research Center for Advanced Medicine, Iuliu Hatieganu University of Medicine and Pharmacy, 400337 Cluj-Napoca, Romania; bobe.petrushev@gmail.com; 5Department of Pathology, Octavian Fodor Regional Institute of Gastroenterology and Hepatology, 400158 Cluj-Napoca, Romania; irus@yahoo.com; 6Department of Hematology, Ion Chiricuta Clinical Cancer Center, 400015 Cluj-Napoca, Romania; deli_dima@yahoo.com; 7Department of Gastroenterology, Octavian Fodor Regional Institute of Gastroenterology and Hepatology, 400158 Cluj-Napoca, Romania; drhstefanescu@gmail.com; 8Department of Gastroenterology, Iuliu Hatieganu University of Medicine and Pharmacy, 400349 Cluj-Napoca, Romania; 9Faculty of Medicine, Grigore T. Popa University of Medicine and Pharmacy, 700115 Iasi, Romania; otiliafrasinariu@gmail.com; 10Intensive Care Unit, Octavian Fodor Regional Institute of Gastroenterology and Hepatology, 400349 Cluj-Napoca, Romania

**Keywords:** hemophagocytic lymphohistiocytosis, ixazomib, multiple myeloma

## Abstract

Hemophagocytic lymphohistiocytosis (HLH) is a rare, elusive, and life-threatening condition that is characterized by the pathologic and uncontrolled secondary activation of the cytotoxic T-cells, natural killer cells (NK-cells), and macrophages of the innate immune system. This condition can develop in sporadic or familial contexts associated with hematological malignancies, as a paraneoplastic syndrome, or linked to an infection related to immune system deficiency. This leads to the systemic inflammation responsible for the overall clinical manifestations. Diagnosis should be thorough, and treatment should be initiated as soon as possible. In the current manuscript, we focus on classifying the HLH spectrum, describing the pathophysiology and the tools needed to search for and correctly identify HLH, and the current therapeutic opportunities. We also present the first case of a multiple myeloma patient that developed HLH following therapy with the ixazomib-lenalidomide-dexamethasone protocol.

## 1. Background and Nomenclature

Hemophagocytic lymphohistiocytosis (HLH) is a rare, elusive, and life-threatening condition, that is characterized by the pathologic and uncontrolled secondary activation of the cytotoxic T-cells, natural killer cells (NK-cells), and macrophages of the innate immune system. Different pathways lead to systemic inflammation due to the increased production of cytokines, including IFN-γ, IL-6, TNFα, IL-1β, and IL-18. This, in turn, is associated with renal, cardiac and hepatic dysfunction, followed by multiple organ failure [1]. The lack of recognition of the disease averts optimal treatment initiation and contributes to increased mortality, with survival ranging between 22% and 55% at 5 years after diagnosis, according to newer studies [2,3].

It is important to raise awareness among medical practitioners regarding the features of HLH, which sadly lacks pathognomonic clinical manifestations or laboratory-specific changes, requiring a combination of clinical and laboratory findings to reach a final diagnosis. When possible, intensive care physicians should be involved in the early management of such patients due to the high risk of organ failure and treatment-associated challenges. A literature review that included 775 patients with adult HLH reported a mean age of HLH onset of 49 years (63% being male), but this is a heterogeneous disease that can be diagnosed at any time from birth to old age [4].

To simplify the condition, patients are categorized as having either primary HLH (familial/genetic mutations affecting lymphocyte cytotoxicity and immune regulation, often encountered in children) or secondary HLH, also known as sporadic HLH (acquired immune dysfunction), which is most frequent in adults triggered by infections or malignancies. When HLH is caused by autoimmune disorders, such as Still’s disease, lupus, or vasculitis, it is called macrophage activation syndrome (MAS or MAS-HLH) [5,6] (Table 1).

The North American Consortium for Histiocytosis (NACHO) has proposed a different nomenclature of the disease based on the obscure primary/secondary classifications described above [12]. The definitions of the spectrum according to NACHO are as stated:(1)HLH syndrome refers to patients who meet consensus diagnostic criteria. The immune activation is often associated with lymphocyte cytotoxicity due to genetic defects;(2)HLH disease: an HLH syndrome in which the distinctive immune dysregulation is the core problem and in which immune suppression could be of benefit. HLH disease may be associated with specific genetic and/or environmental causes.HLH disease can be:-Familial HLH with clear genetic etiology (F-HLH);-HLH associated with malignancy (M-HLH);-HLH associated with rheumatologic conditions (MAS) (R-HLH);-HLH observed after immune-activating therapies (iatrogenic HLH, also called cytokine release syndrome) (Rx-HLH);-HLH associated with immune compromise (either primary immune deficiency or treatment-related immune suppression) (IC-HLH);-HLH not associated with other specific conditions (those with negative or ambiguous genetic findings, with or without infectious triggers) (HLH-disease NOS).(3)HLH disease mimics: disorders that follow the same pattern as HLH syndrome but are caused by other conditions and which might not benefit from immune suppression [12].

In conclusion, the most important aspect to note from the above information is the fact that diagnosis should be based on searching the HLH pattern in patients while meticulously eliminating HLH disease mimics. This will lead to the prompt introduction of supportive treatment as soon as possible, but also takes into account the fact that discovering an HLH pattern might not be enough to initiate HLH-directed therapies. Moreover, HLH disease and HLH disease mimics might overlap on the spectrum, making matters much more complicated.

## 2. Pathogenesis

Primary and secondary HLH, including MAS-HLH, are hyperferritinemia hyperinflammatory syndromes with a common final pathway and different pathogenetic origins [6].

The mechanisms of pathogenesis regarding cellular and molecular pathways are complex. The immune system does not target self-antigens, such as in autoimmune disorders. The main key to the pathogenesis of HLH is represented by a failure of the immune system to diminish the stimulatory effects of various triggers, leading to the uncontrolled activation and proliferation of CD8+ cytotoxic T cells and macrophages. This is followed by a cytokine storm which is responsible for generalized tissue damage and subsequent multiorgan failure [13]. Mutations in some genes alter the normal pathway and result in comparable clinical phenotypes, with altered lymphocyte activation and survival and impaired CD8+ cytotoxic T cells and NK cell-mediated cytotoxicity [5]. For a pictographic understanding of this disease, we have generated Figure 1 and Figure 2.

The usual elimination pathway of tumor cells, infected cells, or antigen-presenting cells (APCs) by NK cells and CD8+ cytotoxic T cells occurs through perforin-dependent cytotoxicity by forming an immunologic synapse. This create a channel/pore through the membrane of the cell/APCs through which cytolytic granules, such as granzyme B, are transferred, and the apoptosis is set in motion [1]. The activation of CD8+ cytotoxic T cells promotes the trafficking of cytotoxic granules from the Golgi apparatus to the cell membrane. These granules dock at the cell membrane, become primed, and fuse with the cell membrane. All these processes are mediated by Rab27a, Munc13-4, Syntaxin-11, and Munc18-2, and result in the release of the cytotoxic mediators (granzyme and perforin) aimed at the target cells [14]. The granule-lining protein CD107a is exposed on the surface of the CD8+ cytotoxic T cells and can serve as a measure of cytotoxic mediators using flowcytometry [14,15] (Figure 1).

Mutations occur in genes such as PRF1, which encodes perforin [5,16,17], UNC13D, which encodes Munc13-4 and is critical for the release of cytolytic vesicles [18,19], STX11 (Syntaxin-11) [20] and STXP2 (Munc18-2), which are involved in cytolysis [21,22], and RAB27A (Rab27a) [23,24], which are responsible for familial HLH forms [7]. Gene defects in XIAP or NLRC4 give rise to HLH cases associated with inflammasome activation [25,26]. Whole-exome sequencing could help identify other potentially HLH-associated genes which could trigger different approaches to this disease [27].

These genetic abnormalities interfere with trafficking, docking, priming for exocytosis, the membrane fusion of cytolytic granules, and the loss of perforin, with the inability to pass these cytolytic granules into the target cell leading to persistent antigens [28].

Going further, the activation of macrophages and the defective killing mechanism of CD8+ T cells involving the APCs pathway augments systemic inflammation by producing more cytokines, including IL-1β, IL-2, IL-6, IL-12, IL-16, IL-18, TNF-α, and IFN-γ [29,30]. Especially, the sustained release of IFN-γ by CD8+ T, which is basically the key cytokine, is responsible for the amplification loops. Besides the activation of the macrophages/monocytes by the CD8+ T cells, tissue injury and inflammation also contribute to the activation of these cells by way of IL1-β and IL-33 [5,31]. Macrophages release IL-1, IL-6, IL-12, and IL-18, with the last two increasing the production of INF-γ by CD8+ T cells, which makes way for a vicious circle [32]. INF-γ stimulates hemophagocytosis (platelets, white blood cells, red blood cells) by the macrophages which release ferritin (Figure 2). Ferritin deposits can be seen in biopsies of tissues such as bone marrow, liver, spleen, and the lymph nodes [33]. The cytokine storm is generated by INF-γ, the chemokine CXCL9, TNF-α, IL-6, IL-10, IL-12, soluble IL-2receptor (CD25), and IL-18, which are responsible for multiorgan failure [7]. Very high levels of IL-18 could help in differentiating MAS from HLH and other autoimmune disorders, which might explain different pathophysiological pathways [34]. Future research might focus on the implications of Toll-like receptors (TLRs) that play a role in the body’s response to inflammation/sepsis and also, apparently, to genetic HLH [35].

## 3. Clinical Manifestations

Due to the cytokines involved (INF-γ, IL-1β, TNF-α, IL-6, IL-12, IL-16, and IL-18), the signs and symptoms of HLH correspond to those of systemic inflammation [29]. Unfortunately, there are no pathognomonic clinical signs that can help in diagnosis, and poorer outcomes have been observed in patients with a higher concentration of cytokines in the blood [1]. Patients can present with recurrent fever [36], neurologic manifestations (seizures, coma), acute respiratory failure due to acute respiratory distress syndrome, bone marrow failure with subsequent cytopenia and increased risk of bleeding, hepato-splenomegaly, elevated liver enzymes with liver dysfunction, or even liver failure, coagulopathy, disseminated intravascular coagulation, skin manifestations (rash, purpura, petechiae), lymphadenopathy (due to the sequestration of lymphocytes), or a sepsis-like syndrome that may progress to multiple organ failure [4,28].

The threshold of treatment initiation should be low due to the heterogenicity of clinical presentation. Therefore, the involvement of multiple organ systems (neurologic, respiratory, cardiovascular, liver, renal, hematologic) will require early intensive care unit (ICU) admission where mechanical ventilation, complex hemodynamic monitoring, and renal replacement therapies can be initiated for organ support.

It is also worth mentioning that there are differences between the adult and pediatric populations regarding the predominance of some of these clinical manifestations. For example, hepatomegaly is found in 95% of children, whereas in adults the range is between 18 and 67%. High levels of ferritin are not so specific for HLH encountered in adults due to the presence of other inflammatory conditions, with hematologic malignancies being one of the most common causes of HLH in adults [37].

## 4. Diagnosis and Differential Diagnosis

In order to establish a certain diagnosis of HLH, an array of criteria must be met (five out of eight). The HLH-2004 diagnostic criteria from the Histiocyte Society are presented in Table 2 [38]. These criteria were developed for children, and are not yet validated for adults. HLH is considered a disease with many faces, because the diagnostic criteria do not include all of the clinical or laboratory features of patients presenting with HLH. Timely diagnosis is mandatory for initiating treatment, improving the quality of life, and decreasing morbidity and mortality.

Complete blood count (CBC), coagulation studies, liver function tests, renal function tests, ferritin levels, triglycerides levels, bone marrow evaluation, and soluble IL-2 receptor levels followed by more complex immunologic and genetic studies should be performed in cases with a high suspicion of the disease [28]. In cases where there is central nervous system (CNS) involvement, neurologic examination followed by brain MRI and lumbar puncture should be performed [40].

When the liver becomes involved, hyperbilirubinemia with jaundice, hepatomegaly, hepatitis, elevated LDH levels, and elevated D-dimer levels can be found. Liver failure could be the final step in disease evolution, and may even require liver transplantation [41]. Coagulopathy with disseminated intravascular coagulation can quickly follow [42]. Liver biopsy can reveal chronic hepatitis with periportal lymphocytic infiltration [43]. Pleocytosis, elevated protein levels and hemophagocytosis can be found in the cerebrospinal fluid in 50% of patients [12]. Cytopenias are encountered in over 80% of patients at the moment of evaluation [44].

Ferritin levels seen in HLH in adults are often high, between 7000 and 10,000 µg/L. Rarely, this figure can be >100,000 μg/L, but elevated levels are neither sensitive nor specific for HLH in adults [45]. Ferritin levels > 10,000 µg/L are >90% sensitive and 98% specific for HLH in children [46], but this is still not enough for a definitive diagnosis. Hyperferritinemia is an indicator of macrophage activation and reflects a high generation of TNF-α [47]. The enhancement of ferroportin-mediated iron efflux by growth differentiation factor 15 (GDF15) appears to be involved in the modulation of iron homeostasis and the development of hyperferritinemia in HLH [48].

T-cell activation is central to the pathophysiology of HLH, meaning that sCD25 levels should be high in those without treatment [49].

HScore is a scoring system that helps in the calculation of the probability of HLH disease. Immunosuppression, fever, organomegaly, triglyceride levels, ferritin levels, alanine aminotransferase, fibrinogen, cytopenia, and hemophagocytosis on bone marrow aspirate are the features that comprise the score. Having an HScore of ≥250 results in a probability >99% of HLH disease, whereas a score of ≤90 corresponds to a <1% probability of HLH disease [50].

To make the matter more challenging, there is the important task of ascertaining different diseases that present in similar ways. The clinical phenotype presentation can be difficult to distinguish from other systemic inflammatory disorders such as sepsis, septic shock, acute liver failure, or multiple organ dysfunction syndrome (MODS) [51]. As such, swift sepsis screening should be performed because, as mentioned before, an infection can also be the trigger of HLH. Nonetheless, in sepsis, there should not be any evidence of T-cell activation, and ferritin levels tend to remain rather constant [28]. SARS-CoV-2 infection can also be associated with an HLH-like syndrome [52].

Transfusion-associated graft-versus-host disease is a complication of non-irradiated blood transfusion, usually after hematopoietic cell transplantation (HSCT), in which donor lymphocytes mount an immune response against the recipient’s tissues, especially the skin, gastrointestinal tract, or bone marrow [53]. In instances when there is difficulty in the differential diagnosis between the two diseases, skin biopsy could help in further interpretation [53,54].

Apparently, elevated levels of the key clinical cytokines involved, such as INF-γ and IL-10 with only moderately elevated IL-6 levels, have good diagnostic accuracy for HLH and could be used for the monitoring of patients and differential diagnosis [55].

Molecular studies regarding HLH-associated proteins (perforin/granzyme B protein expression, SLAM-associated proteins, the X-linked inhibition of apoptosis protein (XIAP), measurements of surface CD107a exposure) can also aid in the diagnosis of HLH [38].

Novel immunotherapies introduced in the fields of oncology and hematology are capable of inducing an HLH-like cytokine storm. The ones that are worth specifying are chimeric antigen receptor-modified T-cells (CAR-T cells), immune checkpoint inhibitors, and bi-specific T-cell engagers such as blinatumomab used for the treatment of B-acute lymphoblastic leukemia [6,56].

As we have mentioned before, in patients with hematological malignancies, HLH can occur as a paraneoplastic syndrome, or because of an infection that occurs in the background of immune deficiency. Of all the malignancies, it seems that hematologic malignancies are most frequently associated with HLH, with an estimated 1% of all lymphoma, myeloma, and leukemia patients being associated with HLH [57]. This might be because these diseases occur from the cells of the immune system. They are not only more prone to inducing immune deficiency, but there is also the possibility for the malignant clone to maintain certain communication pathways with the normal immune cells, influencing their activation [4,58,59]. This is also supported by the fact that HLH occurs more frequently in association with T-cell malignancies than with B-cell malignancies, although the latter has a higher prevalence than the former. This effect might occur due to the role that T-cells normally have in orchestrating the immune system [58]. Communication between the malignant clone and the immune system cells might occur through soluble factors, such as interleukins or microvesicles [1]. Additionally, treatments used in the hematology unit, as well as viral infections which are rare in the normal population, but rather common the immunocompromised population, have also been observed to be associated with HLH [58,60]. Nonetheless, a discussion can be held here, as it is often unknown whether HLH occurs as a paraneoplastic syndrome or a treatment-related condition. The recently approved treatment mentioned above is more likely to induce macrophage activation, and cytokine storms are often represented by CAR-T cells therapies or other T-cell-engaging therapies [56,61,62].

## 5. Current Treatment Strategies and Future Approaches

It is mandatory to conduct a rapid therapeutic approach before the results of the diagnostic or genetic studies are returned, due to the possibility of swift organ dysfunction. This involves taking into consideration the removal of triggers one by one—especially the presence of possible infections and malignancies—and focusing on establishing a prompt diagnosis of the syndrome. Therefore, the initial treatment of primary and secondary HLH disease is the same because, in general, genetic studies are not available at the moment of patient presentation.

In primary HLH, the aim is to suppress the immune response and limit damages that arise because of this effect. The current treatment approach is based on the HLH-94 protocol, 2018 recommendations and HLH-2004 studies, due to the lack of specific therapy. This incorporates an 8-week induction immuno-chemotherapy with dexamethasone, etoposide, and intrathecal methotrexate (when CNS is involved) which aims to reduce the generalized inflammatory process. The 8-week regimen consists of a 2-week phase of daily dexamethasone (10 mg/m^2^) with a twice-weekly administration of etoposide 150 mg/m^2^, followed by the weekly consumption of etoposide for the next 6 weeks, with a reduction in steroid doses [63]. Cyclosporine A can be used to counter possible reactivation. These agents target cells such as lymphocytes, macrophages and the APC [3]. After the 8-week treatment, two pathways can be taken: either weaning off the initial therapy or the continuation of therapy (etoposide 150 mg/m^2^ every 2 weeks and dexamethasone pulses of 10 mg/m^2^/day for 3 days, every 2 weeks) as a bridge to transplantation. Etoposide is cleared by both renal and hepatic routes, with the need to reduce the dosage in case of organ failure [40,63].

Following the treatment initiation, clinical and biological monitoring is fulfilled in order to survey the response to therapy and observe for possible toxicities. The measurement of sCD25 levels can help in guiding therapy and evaluating immune activation and the need for therapy augmentation. Ferritin levels may not be accurate in the treatment response due to slow normalization [12].

In cases of disease relapse or the lack of partial resolution after 2–3 weeks of treatment, salvage therapy should be sought. Where CNS is involved, weekly intrathecal methotrexate and hydrocortisone should be administered, with close observation paid to the resolution of symptoms and cerebrospinal fluid properties [6]. Physicians should provide continuous supportive care for the organs, nursing, screening for opportunistic infection, and blood component administration.

Salvage therapies for refractory HLH consist of second-line therapies such as monoclonal antibodies (alemtuzumab (antiCD52), daclizumab, infliximab, ruxolitinib [64]), anti-thymocyte globulin, or vincristine [65].

Following continuation therapy where the remission phase has been achieved, HSCT should be sought as definitive treatment, especially in cases of familial HLH, cases of progressive or refractory forms of the disease, or where CNS is involved [66]. The conditioning regimen for allogeneic HSCT is realized with cyclophosphamide/fludarabine, busulfan, and etoposide [67].

In the cases of MAS, which is a form of HLH associated with autoimmune disease, the treatment consists of high-dose corticosteroid or cyclosporine A in refractory cases, together with specific therapy for the underlying autoimmune disorder [68].

As for future treatment opportunities, cytokines such as INF-γ could be used as targets (e.g., emapalumab) for monoclonal antibody agents that bind to INF-γ and neutralize it [69]. Alternatively, gene therapies [70] and many other immunotherapies could be used in clinical practice after confirmation across different trials.

## 6. Prognosis

In cases of untreated primary HLH disease, survival rate is around two months, essentially because of the development of multiorgan failure [71]. The HLH 2004 study reported that, in children with familial HLH who were treated according to this protocol, the 5-year survival rate was 59% [44]. A publication from 2011 estimated a 5-year probability of survival of 54 ± 6% (median follow-up of 6.2 years). In patients who received HSCT, the 5-year survival rate was 66 ± 8%. It is worth mentioning that all patients with familial HLH who did not benefit from a transplant died [72]. Another multicenter study has reported that median survival after allogeneic HSCT is 21.5 months [73]. Secondary HLH disease that fails to receive proper treatment has a mortality rate of around 50–75% [40].

## 7. Case Report

In the current manuscript, we want to present the case of a multiple myeloma patient that developed HLH following therapy with ixazomib-lenalidomide-dexamethasone protocol.

A 64-year-old gentleman presented with thrombocytopenia at a routine CBC. Bone marrow aspiration revealed the presence of normal megakaryocyte lineage development, but also showed a plasma cell infiltrate. We also performed immunoglobulin dosing and serum protein electrophoresis, together with immunofixation. Both of these techniques revealed a monoclonal increase in the IgA and lambda light chains. The radiologic assessment revealed the presence of osteolytic lesions at the level of the neurocranium. The patient was diagnosed with stage I multiple myeloma and started therapy with bortezomib-cyclophosphamide-dexamethasone. We considered a cycle of therapy as 21 days. Our initial intent was to follow this approach with autologous HSCT. After three cycles of therapy, we performed another bone marrow aspiration which revealed the persistence of the multiple myeloma clone. As such, we decided to switch the treatment to ixazomib-lenalidomide-dexamethasone protocol, according to Moreau et al. [74].

After a month, the patient subjectively reported fatigability and, at the clinical examination, we observed the presence of erythema at the level of the thorax and the superior limbs. The symptomatology was reduced in intensity following symptomatic therapy. Following cycle 3 of therapy with ixazomib-lenalidomide-dexamethasone, the patient presented with marked fatigability, thrombocytopenia, hypofibrinogenemia, coagulopathy, hyperferritinemia, and jaundice, for which the gastroenterology unit was consulted. A clinical diagnosis of acute liver failure was made, and the liver biopsy was concluded as hepatic HLH (Figure 3).

The patient was transferred to the ICU and started treatment with etoposide, high-dose dexamethasone, and hemodiafiltration, according to a previously published protocol [56]. Unfortunately, the patient died three days later.

## 8. Discussions

In the present manuscript, we presented the case of a multiple myeloma patient that developed HLH following therapy with an ixazomib-based regimen. The main differential diagnosis was whether HLH was caused by the multiple myeloma occurring as a paraneoplastic syndrome or because of the treatment used. Nonetheless, neither possibility can be excluded.

HLH associated with multiple myeloma has been reported in the literature, although on rare occasions, with the team of Dimopoulos et al. presenting a case of multiple myeloma presenting upfront with HLH following chemotherapy [75]. The diagnosis of HLH in association with a treatment regimen based on proteasome inhibitors, immunomodulatory drugs and dexamethasone is very rare, taking into consideration that dexamethasone is subsequently used in the management of HLH. Thus, this is the first case of ixazomib-associated HLH. This being said, the interference of these agents with the immune system does not completely exclude their potential role in the development of HLH.

Other possible interpretations suggest either the presence of primary HLH, or that it was caused by a viral infection. Primary HLH would be unlikely because this entity generally manifests in children, and the vicious presentation that was observed in our patient would be unlikely to have not occurred until his sixties [59], although molecular defects in HLH-associated genes have been reported in patients as old as 62 [76]. Considering that there are cases of HLH associated with viral infections, there is also a possibility that, in this case, a viral infection that triggered HLH could have been missed, as viral screening panels are not commonly used when treating multiple myeloma patients, especially with regimens that should not generally be associated with atypical viral infections [59]. Nonetheless, clinically and laboratory-wise, we did not observe the presence of any viral infection, so the remaining possibility could be represented by a sub-clinical form of viral infection.

## 9. Conclusions

Hemophagocytic lymphohistiocytosis (HLH) is a rare, elusive, life-threatening immune dysregulation condition, with complex pathogenesis that is characterized by the secondary activation of the innate immune system cells. Raising awareness among physicians regarding the existence of this disease is important due to the necessity of early diagnosis for optimal treatment initiation. In the current manuscript, we presented a case of a multiple myeloma patient who developed HLH following therapy with an ixazomib-based regimen. This is worth mentioning because the immunotherapies available for different diseases might induce HLH. Differential diagnoses of HLH should be further investigated in likewise cases. Regarding nomenclature, physicians should try to adopt the newer NACHO classification of HLH disease for the sake of clarity and to assist in identifying disease-specific backgrounds. Future research should focus on creating a deeper understanding of the pathophysiology of the disease, developing precise and redefined diagnostic criteria with better availability of genetic and molecular studies, and pushing the boundaries of treatment with novel immunotherapies which can improve survival and limit the evolution of this deadly syndrome.

## Figures and Tables

**Figure 1 jpm-12-00678-f001:**
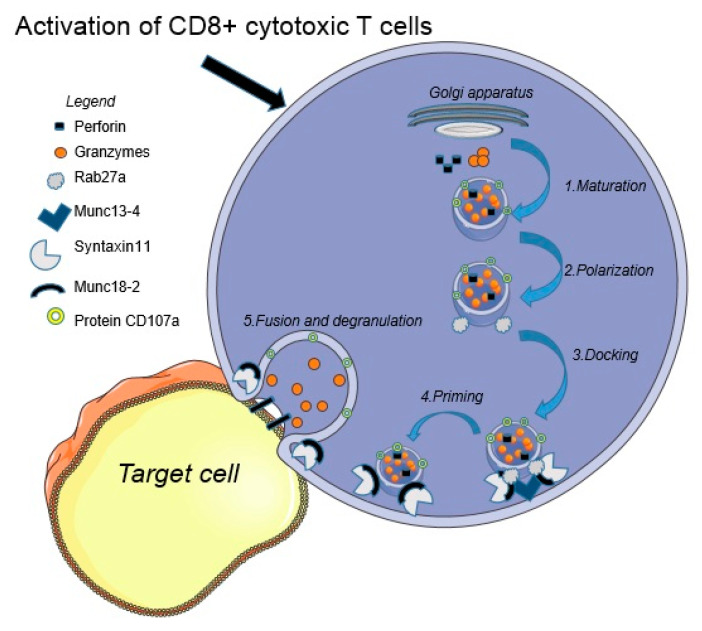
Activation of CD8+ cytotoxic T cells and the pathway of granule and perforin-dependent cytotoxicity. This pathway is formed from the following steps: (1) Maturation: vesicles secreted from the Golgi apparatus are loaded with granzymes and perforins. (2) Polarization: the vesicles are then polarized using Rab27a, a protein important to the subsequent docking of vesicles to the inner side of the cell membrane. (3) Docking: Rab27a from the structure of the vesicles interacts with the inner portion of the cell membrane via a complex formed by Munc13-4, Syntaxin11 and Munc18-2. (4) Priming: the vesicle is prepared for membrane fusion via Syntaxin11 and Munc18-2. (5) Fusion and degranulation: the vesicle and cell membrane fuse and the contents of the vesicle are put into contact with the target cell, forming pores in the membrane of the target cell.

**Figure 2 jpm-12-00678-f002:**
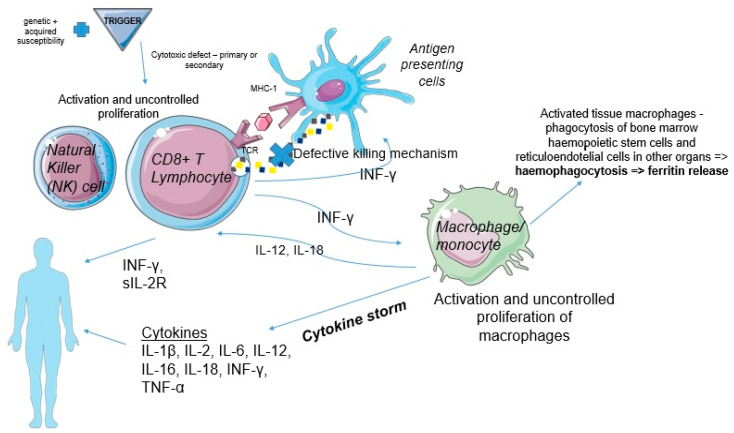
Pathogenesis of HLH. The macrophages become active and see uncontrolled proliferation through various mechanisms. This causes them to release a variety of pro-inflammatory cytokines which are involved in the further amplification effects of HLH. Of note, we can mention INF- γ, IL-1 and IL-6, which are known pro-inflammatory cytokines. Moreover, through IL-12 and IL-18, macrophages also have an effect on cytotoxic T-lymphocytes, activating them and creating a vicious cycle.

**Figure 3 jpm-12-00678-f003:**
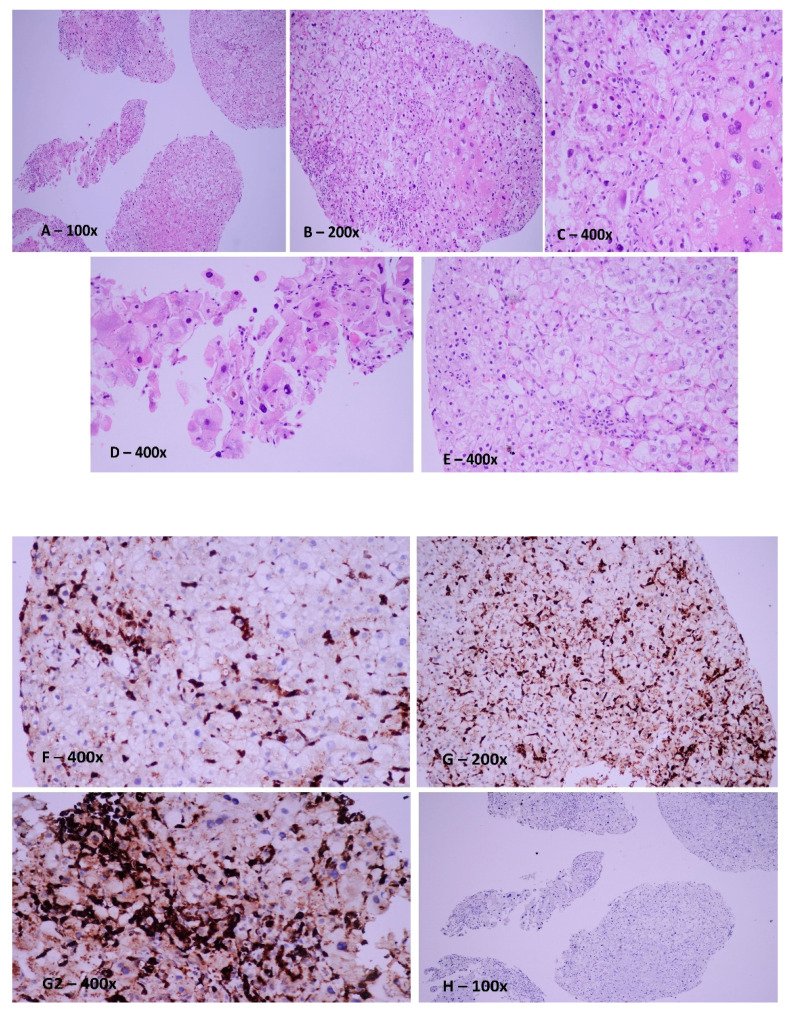
Transjugular liver biopsy–usual Hematoxylin-Eosin staining (Figures (**A**–**E**,**H**)). Dissociated cylindrical biopsy fragments are represented by hepatic parenchyma (image (**A**)) with preserved architecture but with regenerative changes of hepatocytes (image (**B**)) and focal necroinflammatory lobular activity (image (**C**)). There are hyperplastic Kupffer cells in the sinusoidal spaces that are positive in immunohistochemical staining for CD 68 (image (**G**,**G2**)) and negative in an immunoassay for CD 1a. Phagocytosis, intracytoplasmic erythrocytes (image (**D**)), but also lymphocytes (image (**E**)), better identifiable in immunohistochemical staining for CD 68 (image (**F**)), were found in isolation.

**Table 1 jpm-12-00678-t001:** Causes of primary and secondary HLH [6].

Primary HLH (Mendelian Inherited Conditions) [7]	Secondary HLH
Defects in the cytolytic function of cytotoxic T cells and/or NK cells	Infections (EBV, HIV, CMV, SARS-CoV-2, bacterial, fungi, parasites) [8]
Defects in inflammasome regulation	Malignancies (lymphomas) T-cell or natural killer (NK) cell lymphomas, B-cell lymphomas, leukemias, Hodgkin lymphoma, solid tumors (all require a meticulous search for the underlying disease) [9]
	MAS or autoimmune disorders: systemic-onset juvenile idiopathic arthritis (sJIA), adult-onset Still’s disease (ASD), vasculitis, systemic lupus erythematosus (LES) [10]
	Organ (kidney) or stem cell transplantation [11]
	Metabolic, surgery, trauma
	Immunosuppression, vaccination, hemodialysis, immune-activating therapy (e.g., CAR-T therapy)
	Pregnancy

**Table 2 jpm-12-00678-t002:** HLH-2004 diagnostic criteria.

The Diagnosis of HLH Can Be Established If Criterion 1 or 2 are Fulfilled.
1. A molecular diagnosis consistent with HLH: pathologic mutations of PRF1, UNC13D, Munc18-2, Rab27a, STX11, SH2D1A, or BIRC4
2. Diagnostic criteria for HLH fulfilled (≥5 of the 8 criteria below)
(1) Fever ≥ 38.5 °C
(2) Splenomegaly
(3) Cytopenias (affecting ≥2 of 3 lineages in the peripheral blood)
Hemoglobin < 9 g/dL (hemoglobin < 10 g/dL in infants < 4 weeks)
Platelets < 100 × 10^3^/mL
Neutrophils < 1 × 10^3^/mL
(4) Hypertriglyceridemia and/or hypofibrinogenemia
Fasting triglycerides ≥ 3.0 mmol/L (i.e., ≥265 mg/dL)
Fibrinogen ≤ 150 mg/dL
(5) Hemophagocytosis in bone marrow, spleen, liver, or lymph nodes with no evidence of malignancy.
(6) Low or no NK cell activity (according to local laboratory reference)
(7) Ferritin ≥ 500 ng/mL
(8) sCD25 (i.e., α-chain soluble IL-2 receptor) ≥ 2400 U/mL correlated with current disease activity [28]
(9) Elevated CXCL9 [39] (not in the original classification criteria)

## Data Availability

All data is available, either analyzed as figures and tables presented in the current manuscript; or as raw data upon request by any external collaborator or reviewer.

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
