# Peer review of "Mechanistic Insights in Hemophagocytic Lymphohistiocytosis: Subsequent Acute Hepatic Failure in a Multiple Myeloma Patient Following Therapy with Ixazomib-Lenalidomide-Dexamethasone"

_jpm, 2022, doi:10.3390/jpm12050678_

Round 1

Reviewer 1 Report

This timely and interesting review by Catalin Constantinescu, et al. summarized the pathogenesis and current diagnosis standards, treatment strategies and disease prognosis of hemophagocytic lymphohistiocytosis (HLH), a rare and life-threatening systemic inflammatory syndrome. The authors also reported a rare case that a multiple myeloma patient developed HLH with ixazomib-lenalidomide-dexamethasone treatment. It is suitable for publications pending clarification and further discussion of the following points:

  1. Figure 1 depicted the activation of CD8+ cytotoxic T cell. However, it lacks a detailed description of the whole process in the figure legend or in the main text, making it hard to understand. Please describe the process step by step in the figure legend. Please also add description for Figure 2. In addition, make sure to refer to the figure when discussing the topic in the main text.
  2. The section of “pathogenesis” is not very well structured. The discussion of each pathogenic pathway is not in depth and kind of sporadic. Please reorganize this section.
  3. Please carefully check all the references and add citations to the right place. For example, “Mutation in genes like PRF1- which encodes perforin, UNC13D – _encodes Munc13-4 - critical for the release of cytolytic vesicles, STX11 (Syntaxin-11) and STXP2 (Munc18-2)- involved in cytolysis, RAB27A (Rab27a), are responsible for familial HLH forms (7). Gene defects in XIAP or NLRC4 give rise to HLH cases associated with inflammasome activation.” The source for each highlighted pathway/gene should be clearly cited.
  4. Please explain any abbreviations when they first occur in the paper. For example, ICU first appeals in “Clinical manifestations”; however, the explanation is in “Case Report”.
  5. Some editing for English language is required throughout the manuscript due to some mistakes. For example, “There is also worth mentioning that there are differences between the adult and pediatric populations...” should be “it is also worth mentioning”. Please remove any unnecessary lay languages and make the manuscript more professional and scientific.

Author Response

Comments and Suggestions for Authors

This timely and interesting review by Catalin Constantinescu, et al. summarized the pathogenesis and current diagnosis standards, treatment strategies and disease prognosis of hemophagocytic lymphohistiocytosis (HLH), a rare and life-threatening systemic inflammatory syndrome. The authors also reported a rare case that a multiple myeloma patient developed HLH with ixazomib-lenalidomide-dexamethasone treatment. It is suitable for publications pending clarification and further discussion of the following points:

  1. Figure 1 depicted the activation of CD8+ cytotoxic T cell. However, it lacks a detailed description of the whole process in the figure legend or in the main text, making it hard to understand. Please describe the process step by step in the figure legend. Please also add description for Figure 2. In addition, make sure to refer to the figure when discussing the topic in the main text.

Thank you for your feedback. We have expanded the description of both Figure 1 and Figure 2. Both of them are mentioned in the main text.

  1. The section of “pathogenesis” is not very well structured. The discussion of each pathogenic pathway is not in depth and kind of sporadic. Please reorganize this section.

Thank you for your feedback we have completely adjusted the “Pathogenesis” section.

  1. Please carefully check all the references and add citations to the right place. For example, “Mutation in genes like PRF1- which encodes perforin, UNC13D – _encodes Munc13-4 - critical for the release of cytolytic vesicles, STX11 (Syntaxin-11) and STXP2 (Munc18-2)- involved in cytolysis, RAB27A (Rab27a), are responsible for familial HLH forms (7). Gene defects in XIAP or NLRC4 give rise to HLH cases associated with inflammasome activation.” The source for each highlighted pathway/gene should be clearly cited.

Thank you for your feedback. We have added the appropriate citations.

  1. Please explain any abbreviations when they first occur in the paper. For example, ICU first appeals in “Clinical manifestations”; however, the explanation is in “Case Report”.

Thank you for your feedback. We corrected this mistake and assessed the rest of the manuscript for similar mistakes.

  1. Some editing for English language is required throughout the manuscript due to some mistakes. For example, “There is also worth mentioning that there are differences between the adult and pediatric populations...” should be “it is also worth mentioning”. Please remove any unnecessary lay languages and make the manuscript more professional and scientific.

Thank you for your feedback. We addressed the segment of text you mentioned and also proofread the rest of the manuscript. Hopefully, the manuscript is more professional now. Of note, several changes were represented by the deletion of certain text fragments, which could not be marked in red (this is just something that the reviewers and editors should be aware about).

Reviewer 2 Report

In this article, the authors reviewed the literature on HLH in both the adult and pediatric populations. However, in my opinion, this review article does not add any new information to HLH. The layout of the article is typical for review articles. As for a review manuscript, prepared for a journal with almost IF = 5.0, 2/3 of the articles cited are older than 5 years! In addition, the genetic / molecular issues are poorly described.
The case report described in point 7 is a new, first case in multiple myeloma setting after the use of a quite new method of therapy. However, I would suggest to the authors preparation of case report to another journal or changing the form of the article from review to oryginal report.
The article requires linguistic proofreading by a native speaker fluent in medical English.

Author Response

Comments and Suggestions for Authors

In this article, the authors reviewed the literature on HLH in both the adult and pediatric populations. However, in my opinion, this review article does not add any new information to HLH. The layout of the article is typical for review articles. As for a review manuscript, prepared for a journal with almost IF = 5.0, 2/3 of the articles cited are older than 5 years! In addition, the genetic / molecular issues are poorly described.

Thank you for your feedback. We have reviewed the genetic and molecular issues. Regarding the time of the citation, it has to be mentioned that HLH is a rather rare condition, making the papers on this subject rather scarce. Thus, we had to use cite older studies. We also included two of the latest available recommendations regarding the disease.

Jordan MB, Allen CE, Greenberg J, Henry M, Hermiston ML, Kumar A, et al. Challenges in the diagnosis of hemophagocytic lymphohistiocytosis: Recommendations from the North American Consortium for Histiocytosis (NACHO). Pediatr Blood Cancer. 2019 Nov;66(11):e27929.

La Rosée P, Horne A, Hines M, von Bahr Greenwood T, Machowicz R, Berliner N, et al. Recommendations for the management of hemophagocytic lymphohistiocytosis in adults. Blood. 2019 Jun 6;133(23):2465–77.

The case report described in point 7 is a new, first case in multiple myeloma setting after the use of a quite new method of therapy. However, I would suggest to the authors preparation of case report to another journal or changing the form of the article from review to oryginal report.

Thank you for your feedback. We understand that this could be presented as a standalone case report. Nonetheless, it is not uncommon for a case report to be presented alongside a review of the respective issue, so that the reader better understands the problem at hand. More than this, to our knowledge, JPM does not accept case reports (Editors please correct us if we are wrong), but, as we really appreciate MDPI as a publishing service and we had good experience with them, we enjoy publishing with them. Thus, this formed another argument for the format of review plus the case report that you currently observe.

The article requires linguistic proofreading by a native speaker fluent in medical English.

Thank you for your feedback. The linguistic issues have been proofread and we hope the medical English is not a concern anymore. Of note, several changes were represented by the deletion of certain text fragments, which could not be marked in red (this is just something that the reviewers and editors should be aware about).

NOTE to EDITOR: In the original submission we have also uploaded some microscopy pictures which were not added in this article format.

Round 2

Reviewer 2 Report

In this version of the article, the authors took my comments into account. I accept them. The only drawback is reading the manuscript in an uncleaned version. Recommendation: acceptance in present form.

This manuscript is a resubmission of an earlier submission. The following is a list of the peer review reports and author responses from that submission.